# `DyCodeEval`: Dynamic Benchmarking of Reasoning Capabilities in Code Large Language Models Under Data Contamination

Simin Chen [1]   Pranav Pusarla [1]   Baishakhi Ray [1]

[1]Columbia University

## Abstract

The rapid advancement of code large language models (Code LLMs) underscores the critical need for effective and transparent benchmarking methods. However, current benchmarking predominantly relies on publicly available, human-created datasets. The widespread use of these static benchmark datasets makes the evaluation process particularly susceptible to data contamination—an unavoidable consequence of the extensive data collection processes employed during LLM training. Existing methods for addressing data contamination typically face significant limitations, including reliance on substantial human effort and difficulty in managing class imbalances. To overcome these challenges, we propose `DyCodeEval`, a novel benchmarking suite specifically designed to evaluate Code LLMs under realistic contamination scenarios. Given an initial seed programming problem, `DyCodeEval` utilizes multiple agents to systematically extract and modify contextual information without changing the core logic, generating semantically equivalent variations. We introduce a dynamic data generation method and conduct extensive empirical studies on two seed datasets involving 18 Code LLMs. The results demonstrate that `DyCodeEval` effectively assesses the reasoning capabilities of Code LLMs under contamination conditions while producing diverse problem variants, thereby ensuring robust and consistent benchmarking outcomes. Our project webpage can be found at this link[1].

[1]Department of Computer Science, Columbia University. Correspondence to: Simin Chen <sc5687@columbia.edu>.

*Proceedings of the $42^{nd}$ International Conference on Machine Learning*, Vancouver, Canada. PMLR 267, 2025. Copyright 2025 by the author(s).

[1]https://codekaleidoscope.github.io/dycodeeval.html

## 1. Introduction

Large language models (LLMs) have demonstrated significant potential as assistant software developers, particularly in code generation (Chen et al., 2021; Guo et al., 2024; Jiang et al., 2024; Di et al., 2024). Consequently, numerous code-focused LLMs have been developed. These models are trained on vast corpora of natural language and programming language data. Once well trained, they can comprehend human instructions and generate the corresponding code snippets.

As diverse model architectures and training algorithms for code LLMs continue to emerge (Vaswani et al., 2017; Shazeer et al., 2017), a key focus in code LLM research is the effective benchmarking of each model's code reasoning capability. Without a standardized and transparent benchmarking suite, assessing these models' performance and driving improvements becomes a significant challenge.

However, existing benchmarking suites for evaluating code LLMs are inadequate due to their static benchmarking schema, which can lead to potential data contamination from unintended data crawling. Research suggests that such contamination may already be present in current LLMs (Chen et al., 2025; Jain et al., 2024; Dong et al., 2024). Although some methods aim to provide contamination-free benchmarking for code LLMs, they still rely on manual efforts. For example, `LiveCodeBench` (Jain et al., 2024) proposes crawling new programming problems from online platforms and benchmarking LLMs based on timestamps, while `PPM` (Chen et al., 2024) attempts to systematize new programming problems by combining manually defined operators. However, these methods have several limitations: (1) *Significant Manual Effort*: These methods still require substantial manual input to create such datasets. For example, `PPM` necessitates manually defining the lambda operator, while `LiveCodeBench` shifts the burden of manual design to question authors on coding platforms. (2) *Imbalanced Semantic Complexity*: The newly generated benchmarking datasets often lack semantic equivalence with the original ones. As a result, when a model performs worse on these benchmarks, it is challenging to determine whether

the lower score reflects diminished model capabilities or increased benchmark complexity. Thus, these new benchmark results fail to provide meaningful guidance for model developers to improve their models effectively.

To address this limitation, rather than manually creating benchmarking datasets with uncertain semantic complexity, we aim to develop an automated method for dynamically evaluating code LLMs. However, designing such a method presents two key challenges: (1) *Generating Semantically Diverse Yet Complexity-Controlled Problems*. The first challenge is how to ensure the generated problems vary in semantics while maintaining controlled complexity. (2) *Providing Comprehensive Benchmarking*. A proper benchmark programming problem must include fine-grained test cases and canonical solutions to rigorously assess correctness.

To address these challenges, we draw inspiration from metamorphic testing (Chen et al., 2018), a widely used approach in software testing to tackle the oracle problem. In our case, we leverage the principles of metamorphic testing to automate comprehensive benchmarking. Specifically, we define a metamorphic relationship for programming problems. A programming problem includes *complexity-related algorithmic abstraction* and *complexity-unrelated context description*. Modifying the *complexity-unrelated context description* alters the problem's semantics without changing its inherent complexity. Building on this relationship, `DyCodeEval` employs LLM-based agents to generate diverse contexts for a seed problem, automatically transforming existing problems into semantically varied yet complexity-preserving versions. Additionally, `DyCodeEval` integrates a validation agent as a probabilistic oracle to verify the correctness and consistency of the newly generated problems, ensuring reliability.

We used `DyCodeEval` to generate new evaluation sets to assess Code LLM performance under both data contamination and real-world benchmarking scenarios. Our key findings are as follows:

1. Our method effectively reflects Code LLMs' reasoning capabilities in a manually crafted contamination environment (§4.2).

2. The performance of some Code LLMs on our dynamic benchmarks degraded significantly, suggesting potential data contamination of these Code LLMs (§4.3).

3. `DyCodeEval` generates semantically diverse programming problems, and its inherent randomness makes the likelihood of generating identical problems extremely low, thereby reducing the risk of data contamination (§4.4).

4. Despite its randomness, `DyCodeEval` consistently produces stable benchmarking results, ensuring reliable evaluation (§4.5).

We summarize our contribution as follows:

- **Novel Problem Characterization.** We identify a limitation in current static benchmarking schemas, as they are insufficient for effectively evaluating modern Code LLMs, especially when data contamination occurs and the model's training process lacks transparency.

- **New Methodology Design**. We propose a novel approach that separates *context* and *algorithm* in programming problems. Building on this concept, we introduce a dynamic benchmarking method, `DyCodeEval`, which generates programming problems for benchmarking without introducing additional complexity to the dataset. This approach mitigates the impact of data contamination, ensuring transparent and reliable benchmarking.

- **Empirical Findings**. We conduct an empirical evaluation of `DyCodeEval`, and the results demonstrate that traditional static benchmarks can create a false sense of accuracy. In contrast, our dynamic benchmarking approach provides consistently reliable results, even under data contamination scenarios. Additionally, `DyCodeEval` generates semantically diverse programming problems while maintaining stable benchmarking results.

## 2. Background & Related Work

### 2.1. Benchmarking Code LLMs

Code LLMs have been widely adopted in various real-world software engineering applications, leading to the development of numerous benchmarks for evaluating their capabilities in code understanding and reasoning (Chen et al., 2021; Li et al., 2024a; Guan et al., 2025; Austin et al., 2021; Chen et al., 2024; Austin et al., 2021; Yu et al., 2024; Jimenez et al., 2024; Ding et al., 2023; Mathai et al., 2024). Among the many tasks designed to assess code reasoning, this work focuses specifically on the task of natural language to code generation and reviews representative benchmarks in this area. HumanEval (Chen et al., 2021) introduced a human-crafted dataset to evaluate the code generation capabilities of large language models. EvalPlus (Liu et al., 2023) later identified the limitations of HumanEval and MBPP—particularly their limited test case coverage—and proposed a more rigorous benchmark. HumanEval-XL (Peng et al., 2024) further extended HumanEval to support multilingual settings. Fig. 1 illustrates an example from HumanEval, a widely used benchmark

| Prompt | Canonical Solution | Test Cases |
|--------|--------------------|-----------|
| ```def has_close_elements(numbers: List[float], threshold: float):
    """
    Check if in given list of numbers, are any two numbers
    closer to each other than given threshold

    >>> has_close_elements([1.0, 2.0, 3.0], 0.5)
    False
    """``` | ```for idx, elem in enumerate(numbers):
    for idx2, elem2 in enumerate(numbers):
        if idx != idx2:
            distance = abs(elem - elem2)
            if distance < threshold:
                return True
return False``` | ```def check(candidate):
    assert candidate([1.0, 3.9, 4.0], 0.3) == True
    assert candidate([1.0, 3.9, 4.0], 0.05) == False
    assert candidate([1.0, 5.9, 5.0], 0.95) == True
.....``` |

*Figure 1.* Benchmark programming problem example

for the natural language to code generation task. Each programming problem typically consists of three components: a prompt, a canonical solution, and a set of test cases. The prompt is first fed into the Code LLM to generate a candidate solution, which is then executed against hidden test cases to evaluate its correctness.

## 2.2. Data Contamination Free Benchmarking

Data contamination has become a significant concern in benchmarking large language models (LLMs) (Brown et al., 2020; Jain et al., 2024; Chen et al., 2025), as it can lead to inflated performance scores and unreliable evaluations. To mitigate this issue, researchers have proposed various contamination-free benchmarking strategies, which can be broadly categorized into three approaches. The first line of work focuses on data protection through encryption and privatization. For instance, Jacovi et al.(Jacovi et al., 2023) and Rajore et al.(Rajore et al., 2024) propose techniques to safeguard benchmark data from being included in LLM training corpora. The second line of research emphasizes timely benchmark updates. LiveBench (White et al., 2024), for example, compiles questions from recent sources such as math competitions held within the past year and regularly updates its dataset. Similarly, LiveCodeBench (Jain et al., 2024) continuously collects new human-authored programming problems from online platforms like LeetCode to maintain freshness and reduce the risk of contamination. The third line of research explores dynamic generation of evaluation sets. DyVal (Zhu et al., 2024a) uses DAG structures to create dynamic benchmarks, TreeEval (Li et al., 2024b) employs high-performing LLMs to generate and evaluate problems via tree planning, and ITD (Inference-Time Decontamination) (Zhu et al., 2024c) identifies and rewrites leaked benchmark samples while preserving their complexity.

## 2.3. LLM as Judgment Agent

Recently, LLMs have become increasingly used as examiners given their capabilities of analyzing large amounts of data and providing unbiased assessments (Bai et al., 2023; Fernandes et al., 2023). This growing trend has gained

interest for two reasons: (1) Enhanced generation of training/testing data (Li et al., 2024b; Liu et al., 2024) (2) Accurate evaluation and comparison of LLM outputs such as in PandaLM (Wang et al., 2024) and DyVal (Zhu et al., 2024b). Additionally, as LLMs have been able to perform remarkbly well on unseen tasks, they offer a faster, equally accurate alternative to human evaluation, (Chiang & yi Lee, 2023).

## 3. Methods: **DyCodeEval**

### 3.1. Design Overview

There are two key challenges in designing a dynamic evaluation schema for benchmarking code LLMs. (1) *Generating Semantically Diverse yet Complexity-Controlled Problems*: There is currently no systematic method for generating programming problems that maintain a consistent complexity level while ensuring semantic diversity. Existing approaches often rely on manual effort, either through predefined rules or domain experts, making them difficult to scale efficiently and incapable of precisely controlling problem complexity. (2) *Ensuring Comprehensive Benchmarking*: To effectively evaluate code LLMs, the generated programming problems must include fine-grained test cases and canonical solutions to rigorously assess correctness.

We draw inspiration from metamorphic testing to generate programming problems using LLMs as agents. Metamorphic testing, widely used in software engineering, defines relationships to address the automatic oracle problem. In our approach, a programming problem prompt consists of two components: *complexity-related algorithmic abstraction* and *complexity-unrelated context description*. Our key metamorphic relationship states that modifying the *complexity-unrelated context description* preserves both the problem's canonical solutions and complexity, enabling controlled problem generation. Additionally, since LLMs are trained on a vast diverse corpus, we can utilize them as agents to suggest relevant and meaningful *complexity-unrelated context descriptions*, further enhancing problem diversity.

The design overview of DyCodeEval is shown in Fig. 2. Given a seed programming problem from existing benchmarks, DyCodeEval generates a semantically different yet

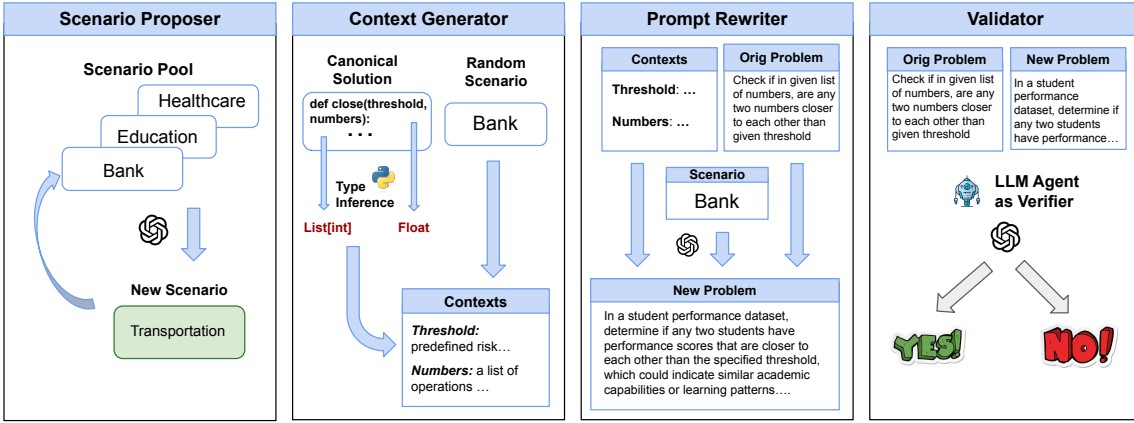

Figure 2. Design overview of `DyCodeEval`

complexity-equivalent problem using a metamorphic relationship. `DyCodeEval` comprises of four agents: (1) Scenario Proposer, (2) Context Generator, (3) Prompt Rewriter, and (4) Validator. The Scenario Proposer suggests real-world domains (e.g., banking, healthcare, education) from which `DyCodeEval` randomly selects one. The Context Generator then analyzes input types in the canonical solution and assigns a relevant context for each input variable based on the selected scenario. The Prompt Rewriter reformulates the problem to align with the input variable contexts and chosen scenario. Finally, the Validator ensures the new problem remains consistent with the original. If inconsistencies are detected, `DyCodeEval` will repeat the aforementioned process until a valid variant is produced.

### 3.2. Detailed Design

**Scenario Proposer Agent.** The Scenario Proposer enhances diversity and minimizes repetition in generated programming problems, reducing potential data contamination. It first selects scenarios from a predefined pool (e.g., banking, healthcare, education, transportation, social networking) and uses them as examples to prompt an LLM for new scenario suggestions. The newly generated scenarios are then added to the pool. By iteratively updating the pool and querying the LLM with varied examples, `DyCodeEval` continuously expands the scenario diversity until the scenario pool reaches a pre-defined size, ensuring the generated scenarios remain diverse and practical. The prompt used for querying the LLM and the suggested scenario examples are listed in Appendix C.

**Context Generation Agent.** After proposing a set of scenarios, the context generation agent randomly selects one from the pool and assigns context information to each input variable of the programming problem based on the chosen scenario.

**Algorithm 1** Type Inference Algorithm. $\texttt{Abstract}(\cdot)$
**Input:** Value list $\mathcal{V}$.
**Output:** Set of data types $\vec{\tau}$.

```
1:  τ⃗ = { }                                    // Initialization.
2:  for each v in 𝒱 do
3:      τ = Type(v)
4:      if τ ∈ Basic Types then
5:          τ⃗ = τ⃗.add(Type(v))
6:      else
7:          τ* = Abstract(ToList(v))
8:          τ⃗.add(τ[τ*])                        // Composite type.
9:      end if
10: end for
11: return τ⃗
```

In languages like Python, input types are not explicitly defined. To address this, the agent uses abstraction for type inference. It analyzes `ASSERT` statements in test cases, collects concrete input values from the canonical solution, and abstracts the input type based on these values. Our type inference algorithm, shown in Alg. 1, works as follows: for each concrete value, it first checks if the type is a basic type (*e.g.,* `int`, `float`). If so, it updates the type set. Otherwise the value is a composite type so it recursively iterates over all the elements and updates the type set with types like `List[int]` or `Tuple[int | string]`. Notice that while our abstract-based type inference may not capture all return value types, it is sound and guarantees that the collected types will always appear in the canonical solution.

After collecting the input data types, the agent prompts the LLM with the scenario and input type information, asking it to assign meaningful context to each input variable based on the given scenario. See Appendix C for prompt templates of our context generation.

**Prompt Rewriting Agent.** With the scenario and context

**Seed Problem:** You will be given a string of words separated by commas or spaces. Your task is to split the string into words and return an array of the words

| Recommendation system | Input: S [string]
User's Blog |

**Generated Problem:** As a content recommendation system developer, you need to process user-generated social media posts to identify relevant keywords for suggesting connections and events. Given a user's post containing words separated by commas or spaces, create a function that breaks down the text into individual words for analysis. The function should handle both comma-separated and space-separated text formats while preserving each distinct word for the recommendation algorithm to process.

*Figure 3.* A generated example from `DyCodeEval`

information for each input variable, the prompt rewriting agent then rewrites the seed programming problem prompt to be tailored to the scenario with meaningful context. Note that we did not ask the LLM to generate the new prompt from scratch. Instead, we provided the detailed scenario and asked it to perform a rewriting task, which is simpler than a generation task. With this approach, leveraging detailed context and a more straightforward task, our agent can generate semantically diverse programming problem prompts. See Appendix C for prompt templates of our prompt rewriting.

**Validation Agent.** Although we provide the LLM with detailed scenario and context information for rewriting, there are cases where the rewriting agent unintentionally alters the consistency. To address this, we design a validation agent to assess whether the generated question maintains the integrity of the original intent and informational content. The validation prompt is designed from two angles: (1) it directs the LLM to compare the seed programming problem prompt with the rephrased prompt, ensuring the preservation of the core concept and factual accuracy, and (2) it asks the LLM to check whether the seed canonical solutions align with the generated programming problem prompt. Specifically, we design two comparison prompts to query the LLM and retain only those rewritten prompts for which both comparison responses are "YES".

To ensure the consistency of the generated programming problems, we also include a human verification step. The details of our validation prompt and the human verification process are presented in Appendix C and Appendix D.

Fig. 3 illustrates an example of programming problems that are semantically diverse yet complexity-equivalent, generated under the scenario of a recommendation system with the context of a user's blog. From this example, we observe that our step-by-step guided approach significantly enhances the semantic diversity of the generated problems, while also reducing the risk of data contamination. This is achieved by leveraging the vast combination space of scenarios and contexts.

### 3.3. Theoretical Collision Analysis

`DyCodeEval` generates programming problems dynamically with randomness, reducing the risk of potential data contamination. To analyze this, we conduct a collision analysis. The randomness in `DyCodeEval` arises from both the scenario proposal and context generation phases. We assume the scenario proposer generates $||\mathcal{S}||$ scenarios, and for each scenario, the context generation produces $||\mathcal{C}||$ contexts, while ignoring randomness in the rewriting phase. We also assume that the random sampling process follows a uniform distribution. Based on this, we present the following theorem.

**Theorem 3.1.** *After running* `DyCodeEval` $M + 1$ *times on the same seed problem, then the probability that the $M$ samples after the first are all different from the first sampled item satisfies:* $P \geq 1 - \exp\left(-\frac{M}{||\mathcal{S}|| \times ||\mathcal{C}|| - 1}\right).$

**Theorem 3.2.** *After running* `DyCodeEval` $M$ *times on the same seed problem, If $M << ||\mathcal{S}|| \times ||\mathcal{C}||$, the probability of at least one collision (i.e., two or more generated problems being the same) after $M$ generations satisfies the following bound:* $P \leq 1 - \exp\left(-\frac{M^2 - M}{2||\mathcal{S}|| \times ||\mathcal{C}||}\right).$

**Theorem 3.3.** *Consider the seed dataset of size $\mathcal{D}$, After running* `DyCodeEval` $M + 1$ *times on this dataset, If $M << ||\mathcal{S}|| \times ||\mathcal{C}||$, then the probability that the $M$ generated datasets after the first one are not the same as the first generated dataset satisfies:* $1 - e^{-\frac{M}{(||\mathcal{S}|| \times ||\mathcal{C}||)^{\mathcal{D}} - 1}} \leq P$

The proof could be found in Appendix A.

## 4. Evaluation

### 4.1. Experimental Setup

**Seed Dataset.** We conduct our evaluation using two datasets: *HumanEval* (Chen et al., 2021) and *MBPP-Sanitized* (Austin et al., 2021). Both datasets are widely utilized in existing research and serve as standard benchmarks for evaluating code generation models. More details about the dataset could be found in Appendix B.

**Implementation Details.** We use CLAUDE-3.5-SONNET as our foundation model to generate the benchmarking dataset. Specifically, we create 50 scenarios, and for each scenario, we randomly generate 50 contexts. During dataset generation, we set the LLM temperature to 0.8, while in our validation agent, we use a temperature of 0. For each code LLM under benchmarking, we employ `vLLM` to launch the model. For closed-source code LLMs, we query the commercial API for evaluation.

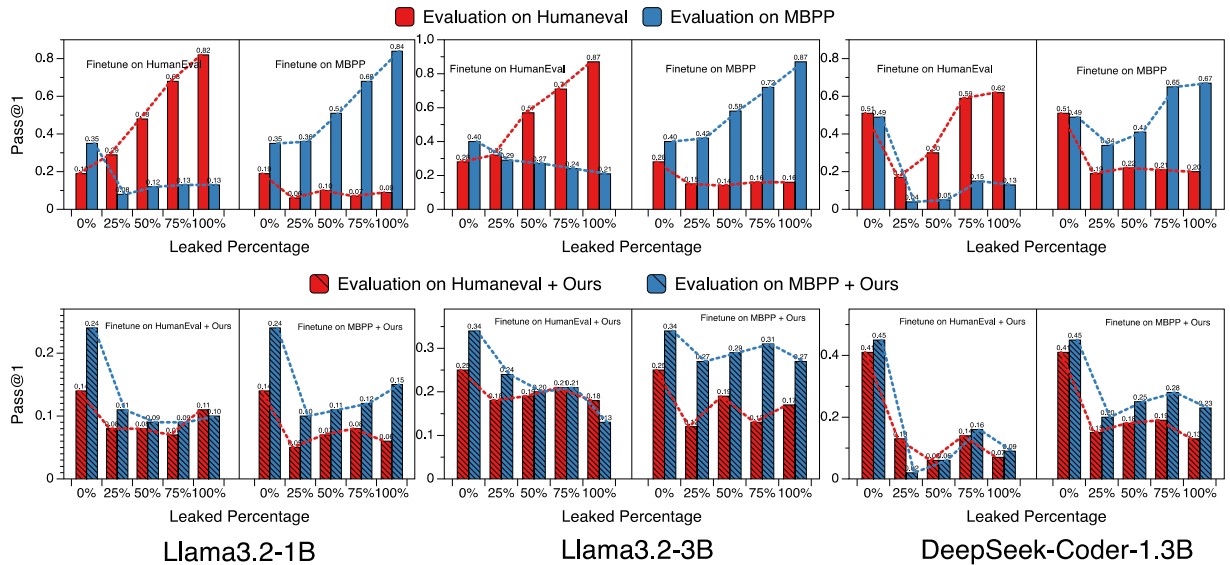

*Figure 4.* Results of benchmarking on contaminated models

## 4.2. Benchmarking Contaminated Model

**Models.** We conduct our study with three public-available Code LLMs: LLAMA-3.2-1B, LLAMA-3.2-3B, and DEEPSEEK-CODER-1.3B.

**Model Contamination Process.** For each model, we simulate data contamination by intentionally leaking a portion of the benchmarking dataset during fine-tuning. We experiment with leaked data percentages of 0%, 25%, 50%, 75%, and 100%, producing four distinct contaminated models. Each polluted model is then evaluated on the benchmarking dataset using the Pass@1 metric. The formal definition of Pass@1 is shown in (1), where $n$ is the number of the generated solution candidate, and $c$ is the number of the correct solutions that can pass all test cases.

$$\texttt{Pass@K} = \mathbb{E}_{\text{Problems}} \left[ 1 - \frac{\binom{n-c}{k}}{\binom{n}{k}} \right] \qquad (1)$$

**Main Results.** The study results are presented in Fig. 4, where there are two rows and three columns. Each column represents evaluation on a different LLM while the rows show static (first) vs dynamic (second) benchmarking. In each column, the left section displays the results for the model fine-tuned on the *HumanEval* dataset, while the right section shows the results for the model fine-tuned on the *MBPP* dataset. The red bars represent the performance of the fine-tuned model benchmarked on the *HumanEval* dataset, and the blue bars represent its performance benchmarked on the *MBPP* dataset.

From the results, we make the following observations: (1)

Data contamination creates a false sense of code reasoning capability under static benchmarks. When the benchmarking dataset is leaked and used for fine-tuning, the model achieves a higher Pass@1 score on the corresponding benchmark. However, this improvement does not accurately reflect the model's true reasoning ability, as its performance declines on other benchmarks that were not included in fine-tuning. (2) Our dynamic benchmarking mitigates the impact of data contamination. Different from static benchmarks, our approach prevents contaminated models from achieving artificially high Pass@1 scores after fine-tuning. This is due to the randomness in our method, which ensures minimal or no overlap between different runs, reducing the risk of direct data leakage. (3) Our dynamic benchmarking dataset provides results comparable to manually curated, non-contaminated datasets. In static benchmarking, as the percentage of leaked data increases, the model's Pass@1 score on the contaminated benchmark steadily improves. However, its performance on other benchmarks remains relatively stable, showing little variation across different contamination levels. Interestingly, this stability also applies to our method. If the base model is not contaminated on the selected seed dataset, this suggests that our approach provides competitive benchmarking results similar to those of human-curated datasets. (4) A notable anomaly is observed in DEEPSEEK-CODER. When only 25% of the benchmarking dataset is used for fine-tuning, the model's Pass@1 score drops below that of the original, unmodified model. We hypothesize that the model may already be overfitted to the contaminated dataset, and further fine-tuning with limited data could destabilize this overfitting without providing enough new information to help the model adapt.

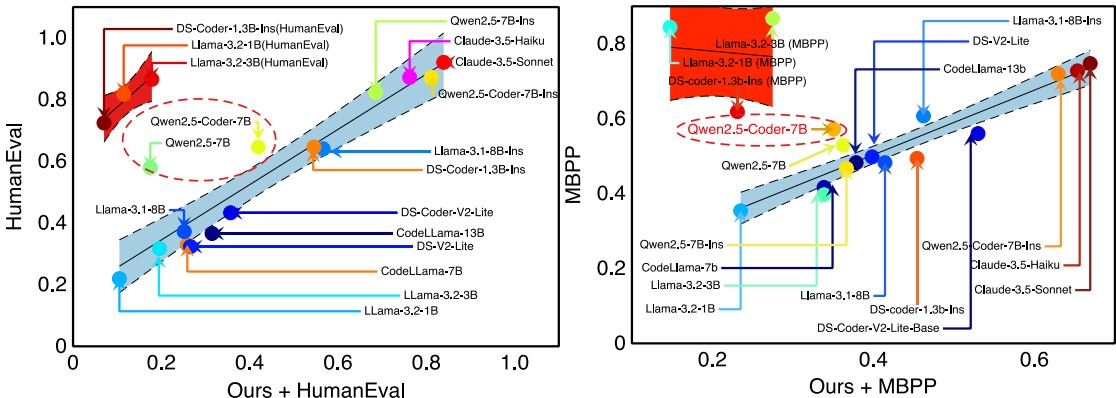

*Figure 5.* The in-the-wild benchmarking results

## 4.3. Benchmarking In-the-Wild Model

We then apply `DyCodeEval` to benchmark more in-the-wild code LLMs, besides the models used in §4.2. We consider the following code LLMs: LLAMA-3.1-8B, CODELLAMA-7B, CODELLAMA-13B, DEEPSEEK-V2-LITE, DEEPSEEK-CODER-V2-LITE-BASE, LLAMA-3.1-8B-INSTRUCT, QWEN2.5-CODER-7B, QWEN2.5-7B-INSTRUCT, QWEN2.5-7B, CLAUDE-3.5-HAIKU, CLAUDE-3.5-SONNET,QWEN2.5-CODER-7B-INSTRUCT .

The results are presented in Fig. 5, with the left figure showing the results on HumanEval and the right showing the results on MBPP. In each figure, the x-axis represents the `Pass@1` scores on our generated dataset, and the y-axis represents the `Pass@1` scores on the seed dataset. The blue region corresponds to the regression area of the in-the-wild model, the red region represents the regression area of the overfitted model on this dataset, and the orange area indicates the overfitted model on the other dataset.

From these results, we observe that for both seed datasets, the in-the-wild model's `Pass@1` scores maintain a linear relationship, while the overfitted model appears as an outlier. A notable finding from our in-the-wild evaluation is that the model QWEN2.5-CODER-7B consistently falls outside the 95% confidence interval of the regression area, suggesting it may be contaminated on both datasets.

## 4.4. Problem Diversity

To evaluate the diversity of the generated programming problems, we conduct two experiments: one for external diversity and one for internal diversity. External diversity quantifies the dissimilarity between the generated and seed problems, while internal diversity measures the diversity within each problem-generation method across trials. We use two metrics: *BLEU-4* to measure syntactical diversity and *cosine similarity* of the prompt's semantic embedding

to measure semantic diversity. For semantic embedding, we use the GPT-2 model to obtain the embedding of each natural language prompt. Moreover, we also consider `PPM` (Chen et al., 2024) and a series of robustness-based mutations (Wang et al., 2023), such as token replacement, insert blank lines, as our comparison baseline.

The diversity results are shown in Table 1, where the first four columns represent internal diversity and the last four columns represent external diversity. From the results, we observe that `DyCodeEval` generates diverse programming problems both syntactically and semantically. Additionally, we find that all baseline methods exhibit high BLEU-4 and semantic similarity scores, as they rely on rule-based approaches to mutate the programming problems, which do not introduce significant diversity. In contrast, `DyCodeEval` leverages an LLM agent to suggest different scenarios and contexts, significantly increasing diversity.

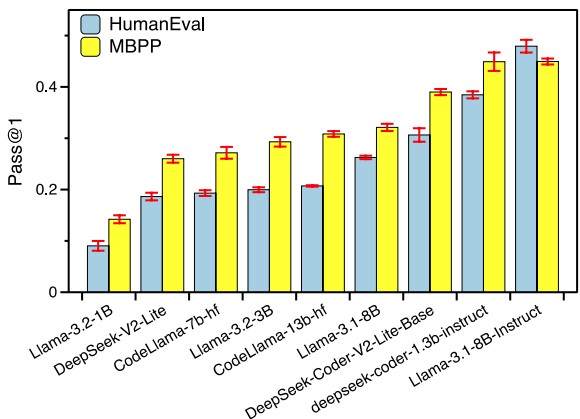

*Figure 6.* Stability results

*Table 1.* Diversity results

| Methods | Internal Diversity | | | | External Diversity | | | |
|---|---|---|---|---|---|---|---|---|
| | HumanEval | | MBPP | | HumanEval | | MBPP | |
| | BLEU-4 ↓ | SemSim ↓ | BLEU-4 ↓ | SemSim ↓ | BLEU-4 ↓ | SemSim ↓ | BLEU-4 ↓ | SemSim ↓ |
| Base | 1.00 | 1.00 | 1.00 | 1.00 | 1.00 | 1.00 | 1.00 | 1.00 |
| Token Mutation | 0.72 | 0.95 | 0.66 | 0.92 | 0.82 | 0.96 | 0.76 | 0.95 |
| Char Mutation | 0.81 | 0.97 | 0.78 | 0.94 | 0.84 | 0.97 | 0.78 | 0.92 |
| Func Mutation | 1.00 | 1.00 | 1.00 | 1.00 | 0.98 | 1.00 | 0.98 | 1.00 |
| Insert Line | 1.00 | 1.00 | 1.00 | 1.00 | 1.00 | 1.00 | 1.00 | 1.00 |
| CommSyntax | 1.00 | 1.00 | 1.00 | 1.00 | 0.81 | 0.98 | 0.73 | 0.99 |
| PPM | 0.97 | 0.96 | 0.96 | 0.94 | 0.69 | 0.89 | 0.57 | 0.84 |
| **Ours** | **0.27** | **0.74** | **0.18** | **0.73** | **0.17** | **0.59** | **0.02** | **0.59** |

## 4.5. Benchmarking Stability

Note that `DyCodeEval` generates a unique benchmarking dataset each time. To assess its stability, we evaluate whether `DyCodeEval` can produce consistent benchmarking results despite this randomness. Specifically, we run `DyCodeEval` 10 times and measure the `Pass@1` scores across these 10 generated benchmark datasets.

The mean and standard deviation of the `Pass@1` scores are presented in Fig. 6. The results show that the variance in benchmarking scores is minimal compared to the mean values, indicating that `DyCodeEval` provides stable benchmarking results across different random trials.

## 4.6. Impact of Foundation LLM

In this section, we evaluate the feasibility of using less advanced LLMs to reduce dataset generation costs. Specifically, we replace our foundation model, CLAUDE-3.5-SONNET, with CLAUDE-3.5-HAIKU. We manually sample and assess generated problems from each model, checking their consistency rate. Our observations show that the consistency rate drops from 95% to 83%, highlighting the need for robust and capable LLMs to serve effectively as foundation models.

## 5. Dynamic Evaluation Metrics

Leveraging the dynamic nature of our method, we propose a new metric, `DyPass`, to address the limitations of the current gold standard, `Pass@K`. Unlike `Pass@K`, which generates $n$ candidate solutions for a fixed problem prompt and evaluates the correctness, our approach creates $n$ semantic prompt variants of a seed problem. These prompt variants preserve the complexity of the original problem by modifying only the description while maintaining the same underlying algorithmic abstraction. Furthermore, $n$ prompt variants expand the input space beyond that of `Pass@K`, making it more challenging to achieve full coverage. As a result, `DyPass` provides a more rigorous assessment of code LLMs' reasoning abilities, particularly under potential

data contamination. Compared to `Pass@K`, which evaluates solutions within a fixed problem context, `DyPass` introduces contextual variations during benchmarking. This allows it to better distinguish whether a model is merely memorizing the problem context or genuinely reasoning to solve it.

*Table 2.* Comparison of `Pass@K` and `DyPass@K` on contaminated Models

| Model | Pass@K | | | DyPass@K | | |
|---|---|---|---|---|---|---|
| | k=3 | k=5 | k=10 | k=3 | k=5 | k=10 |
| Llama-3.2-1B | 0.22 | 0.27 | 0.34 | 0.17 | 0.21 | 0.26 |
| Llama-3.2-1B (C) | 0.82 | 0.83 | 0.85 | 0.13 | 0.15 | 0.17 |
| Llama-3.2-3B | 0.35 | 0.40 | 0.48 | 0.31 | 0.36 | 0.43 |
| Llama-3.2-3B (C) | 0.88 | 0.88 | 0.89 | 0.24 | 0.27 | 0.29 |

*Table 3.* Comparison of `Pass@K` and `DyPass@K` on in-the-wild models

| Model | Pass@K | | | DyPass@K | | |
|---|---|---|---|---|---|---|
| | k=3 | k=5 | k=10 | k=3 | k=5 | k=10 |
| CodeLlama-7b-hf | 0.39 | 0.46 | 0.56 | 0.34 | 0.40 | 0.49 |
| CodeLlama-13b-hf | 0.48 | 0.57 | 0.68 | 0.37 | 0.45 | 0.53 |
| Llama-3.2-1B | 0.22 | 0.27 | 0.34 | 0.17 | 0.21 | 0.26 |
| Llama-3.2-3B | 0.35 | 0.40 | 0.48 | 0.31 | 0.36 | 0.43 |
| Llama-3.1-8B | 0.48 | 0.56 | 0.65 | 0.39 | 0.45 | 0.53 |
| Llama-3.1-8B-Instruct | 0.72 | 0.77 | 0.83 | 0.64 | 0.69 | 0.75 |

To demonstrate the advantages of `DyPass`, we compare it against `Pass@K` on both contaminated and in-the-wild models, with $K = 3, 5, 10$ for evaluation. The results are presented in Table 2 and Table 3. From the results in Table 2, we observe that when the model is trained on leaked data, the static metric `Pass@K` fails to accurately reflect the model's reasoning capabilities, with all `Pass@K` scores rising to very high levels (e.g., from 0.82 to 0.89). In contrast, our dynamic metric `DyPass` @K shows a slight decrease rather than a significant increase, highlighting the sensitivity of `DyPass` to data contamination. When comparing `Pass@K` and `DyPass` @K on models that were not specif-

ically trained on the leaked dataset, both metrics show consistency in benchmarking code LLMs. Based on these observations, we conclude that our dynamic metric, `DyPass`, effectively reflects the reasoning capabilities of code LLMs, even under data contamination. Moreover, `DyPass @K` aligns with static benchmarking metrics when there is no data contamination.

## 6. Conclusion

In this paper, we introduce `DyCodeEval`, a new benchmarking suite that dynamically generates semantically equivalent diverse problems as a way to combat data contamination. We break this generation up into four distinct steps to systematically develop a new programming problem with the same algorithmic complexity but different context. Our experimental results show that while `Pass@k` with current benchmarks have caused inflated model scores, `DyCodeEval`-generated questions with `DivPass` has proven to perform as a reliable evaluation tool. We believe that these results show a promising path forward.

Our proposed work has several limitations: (1) Although LLMs provide a fully automated way to generate diverse programming problems for benchmarking, their computational cost is a significant concern. We found that a very large LLM is required to generate programming problems with a high consistency rate. Therefore, a future improvement could focus on enhancing the efficiency of the problem generation phase. (2) While generating questions using DyCodeEval, we observed instances where excessive information was provided, potentially confusing the reader. This highlights the opportunity for improving prompt generation through further experimentation.

## Acknowledgements

This work was supported in part by CCF 2313055, CCF 2107405, CAREER 2025082, and FAI: 2040961. Any opinions, findings, conclusions, or recommendations expressed herein are those of the authors.

## Impact Statement

Assessing the overall capabilities of large language models (LLMs) is essential for ensuring their reliable and safe deployment in society. However, data contamination can inflate evaluation accuracy, obscuring a model's true performance. To address this, we propose a new benchmarking method, `DyCodeEval`, which enables more accurate measurement of LLM capabilities and provides deeper insights into their behavior.

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

# A. Proof of Theorem

## A.1. Proof of Theorem 3.1

The total number of possible distinct outcomes is $||\mathcal{S}|| \times ||\mathcal{C}||$, the size of the random space, let $N = ||\mathcal{S}|| \times ||\mathcal{C}||$ Since each of the $M$ samples must **not** match $X_1$, and they are drawn independently, the exact probability is:

$$P(X_2 \neq X_1, \ldots, X_{M+1} \neq X_1) = \left( \frac{N-1}{N} \right)^M.$$

We use the standard inequality for the logarithm:

$$\ln(1-x) \geq -\frac{x}{1-x}, \quad \text{for } 0 < x < 1.$$

Applying this to $\frac{1}{N}$, we get:

$$\ln \left( \frac{N-1}{N} \right) = \ln \left( 1 - \frac{1}{N} \right) \geq -\frac{1/N}{1 - 1/N} = -\frac{1}{N-1}.$$

Exponentiating both sides:

$$\frac{N-1}{N} \geq e^{-\frac{1}{N-1}}.$$

Raising both sides to the power $M$:

$$\left( \frac{N-1}{N} \right)^M \geq e^{-\frac{M}{N-1}}.$$

## A.2. Proof of Theorem 3.2

Each sampled item is drawn independently and uniformly from the space of size $N$. We analyze the probability that all $M$ sampled items are distinct.

The first sample can be any of the $N$ items, the second sample must avoid the first one, so there are $N - 1$ choices. Continuing this way, the probability that all $M$ items are distinct is:

$$P(\text{no collisions}) = \frac{N}{N} \times \frac{N-1}{N} \times \frac{N-2}{N} \times \cdots \times \frac{N-(M-1)}{N}.$$

Rewriting in factorial form,

$$P(\text{no collisions}) = \frac{N!}{N^M (N-M)!}.$$

According to our assumption $M << ||\mathcal{S}|| \times ||\mathcal{C}||$, Using the Stirling's approximation, then we have

$$\frac{N!}{(N-M)!} \geq N^M \exp \left( -\frac{M(M-1)}{2N} \right),$$

we get

$$P(\text{no collisions}) \geq \exp \left( -\frac{M(M-1)}{2N} \right).$$

The probability of at least one collision is the complement:

$$P(\text{at least one collision}) = 1 - P(\text{no collisions}).$$

Using the bound we derived,

$$P(\text{at least one collision}) \leq 1 - \exp\left(-\frac{M^2 - M}{2N}\right) = 1 - \exp\left(-\frac{M^2 - M}{2||\mathcal{S}|| \times ||\mathcal{C}||}\right)$$

### A.3. Proof of Theorem 3.3

Each sample can be represented as a $D$-tuple of balls $(b_1, b_2, ..., b_D)$, where each $b_i$ is one of the $N$ balls from bag $i$. The total number of possible sample sets is:

$$T = N^D$$

Since each draw is independent, each sample set is chosen uniformly from $T$, meaning the probability of selecting any specific tuple is:

$$\frac{1}{N^D}$$

Let $X_1$ be the initial sample (first draw). For each subsequent draw $X_i$ (where $i = 2, \ldots, M + 1$), the probability that $X_i = X_1$ (i.e., an exact match) is:

$$P(X_i = X_1) = \frac{1}{N^D}$$

Then Theorem 3.3 could be proved through Theorem 3.1.

## B. Dataset Description.

The *HumanEval* dataset, developed by OpenAI, is an open-source benchmark for evaluating the code generation capabilities of pre-trained code language models (LLMs). It comprises 164 Python programming problems, each consisting of a prompt, a canonical solution, and corresponding test inputs. Each prompt includes a natural language problem description, a function definition, and input/output examples.

The *MBPP-Sanitized* dataset, proposed by Google, features 427 Python programming problems collected through crowd-sourcing. Unlike *HumanEval*, it is a zero-shot dataset, meaning its prompts do not include input/output demonstrations. To enhance its utility in experiments, we refined the prompt format by adding function headers and converting natural language instructions into function docstrings.

## C. Prompt Templates & Scenario Examples

In the following, we show the scenario examples and prompt templates used during the four steps of `DyCodeEval` process.

## C.1. Template for Scenario Proposer Agent

**Prompt for Scenario Proposer Agent**

```
Suggest real-world scenarios that provide meaningful context in the
following areas:  {S₁}, {S₂}, {S₃}, {S₄}, {S₅}, and any other practical
fields.  Each scenario should be general but applicable, providing useful
insight for potential applications.

For clarity, return each scenario on a separate line without additional
explanation.  Use the example below for reference.

Please put your suggested Real-world Scenarios in <scenario></scenario>
tags.

# Scenario Examples:
<example>
{EXAMPLE}
</example>
```

## C.2. Example for Scenario Proposer Agent

**Example for Scenario Proposer Agent**

```
Suggest real-world scenarios that provide meaningful context in the
following areas:  transportation, education, healthcare, banking, social
networking, and any other practical fields.  Each scenario should be general
but applicable, providing useful insight for potential applications.

For clarity, return each scenario on a separate line without additional
explanation.  Use the example below for reference.

Please put your suggested Real-world Scenarios in <scenario></scenario>
tags.

# Scenario Examples:
<example>
Banking - Fraud Detection.
</example>
```

## C.3. Prompt for Context Generator Agent

---

**Prompt for Content Generator Agent**

Given the natural language problem description, input types, and a real-world
scenario. For each variable in the problem, provide a meaningful context tailored
to the given scenario. The context should explain how each variable is involved in
or relates to the scenario, ensuring practical relevance.

Please ensure for to put your meaningful context in <context></context> tags.

# Problem Description:
<problem_description>
**{PROBLEM DESCRIPTION}**
</problem_description>

# Input Types:
<input_types>
**{INPUT VARIABLE TYPES}**
</input_types>

# Real-world Scenario:
<scenario>
**{SCENARIOS}**
</scenario>

# Instructions:
– For each variable in the input types, generate only one context that highlights its
role or significance within the problem and scenario.
– The context should help to clarify the variable's meaning and importance, ensuring
that it fits into the given real-world scenario.
– Provide only the contexts for the variables (no additional reasoning steps).

# Example:
## Problem Description Example:
<problem_description>
Determine if the average temperature in a city exceeds a certain threshold during a
week.
</problem_description>

## Input Types:
<input_types>
temperatures:  list of float
threshold:  float
</input_types>

## Scenario:
<scenario>
Climate Analysis – Monitoring Urban Heat Trends
</scenario>

## Generated Contexts:
<context>
temperatures:  Daily recorded temperatures in a city, analyzed for urban heat
trends.
threshold:  Critical temperature level indicating hazardous or abnormal heat.
</context>

---

## C.4. Example for Context Generator Agent



**Example for Context Generator Agent**

```
Given the natural language problem description, input types, and a real-world
scenario.  For each variable in the problem, provide a meaningful context tailored
to the given scenario.  The context should explain how each variable is involved in
or relates to the scenario, ensuring practical relevance.

Please ensure for to put your meaningful context in <context></context> tags.

# Problem Description:
<problem_description>
You're given a list of deposit and withdrawal operations on a bank account that
starts with zero balance.  Your task is to detect if at any point the balance of
account fallls below zero, and at that point function should return True.  Otherwise
it should return False.
</problem_description>

# Input Types:
<input_types>
operations:  list of int
</input_types>

# Real-world Scenario:
Education - Adaptive Learning Assessment and Skill Gap Analysis
</scenario>

# Instructions:
- For each variable in the input types, generate only one context that highlights its
 role or significance within the problem and scenario.
- The context should help to clarify the variable's meaning and importance, ensuring
 that it fits into the given real-world scenario.
- Provide only the contexts for the variables (no additional reasoning steps).

# Example:
## Problem Description Example:
<problem_description>
Determine if the average temperature in a city exceeds a certain threshold during a
week.
</problem_description>

## Input Types:
<input_types>
temperatures:  list of float
threshold:  float
</input_types>

## Scenario:
<scenario>
Climate Analysis - Monitoring Urban Heat Trends
</scenario>

## Generated Contexts:
<context>
temperatures:  Daily recorded temperatures in a city, analyzed for urban heat
trends.
threshold:  Critical temperature level indicating hazardous or abnormal heat.
</context>
```



## C.5. Prompt for Prompt Rewriter Agent

**Prompt for Prompt Rewriter Agent**

```
Given a seed programming problem description, a selected real-world
scenario, and contextualized input variables, rewrite the original problem
to make it relevant to the scenario.  The rewritten problem should:
- Preserve the original problem's complexity and constraints.
- Ensure the problem remains solvable with the same solution approach.
- Be clear, concise, and logically consistent within the new context.
- Just return the rewritten problem description without any additional
commentary or steps, and do not include any input output demons in your
problem description.
- Limit the new rewritten problem description to 1-3 sentences.
- Make sure your rewritten problem description is clear, concise and
contains no unnecessary information.

Please ensure to put your rewritten problem description in
<new_problem></new_problem> tags.

# Problem Description:
{PROBLEM DESCRIPTION}

# Real-World Scenario:
{SCENARIO}

# Contextualized Input Variables:
{INPUT VARIABLES}
```

## C.6. Example for Prompt Rewriter Agent

**Example for Prompt Rewriter Agent**

```
Given a seed programming problem description, a selected real-world
scenario, and contextualized input variables, rewrite the original problem
to make it relevant to the scenario.  The rewritten problem should:
- Preserve the original problem's complexity and constraints.
- Ensure the problem remains solvable with the same solution approach.
- Be clear, concise, and logically consistent within the new context.
- Just return the rewritten problem description without any additional
commentary or steps, and do not include any input output demons in your
problem description.
- Limit the new rewritten problem description to 1-3 sentences.
- Make sure your rewritten problem description is clear, concise and
contains no unnecessary information.

Please ensure to put your rewritten problem description in
<new_problem></new_problem> tags.

# Problem Description:
You're given a list of deposit and withdrawal operations on a bank account
that starts with zero balance.  Your task is to detect if at any point the
balance of account fallls below zero, and at that point function should
return True.  Otherwise it should return False.

# Real-World Scenario:
Social Networking - Advanced Content Recommendation and User Interest
Matching

# Contextualized Input Variables:
A sequence of user interaction events (deposits/withdrawals) representing
content engagement metrics in a social networking platform, where each
operation tracks how users interact with recommended content, potentially
influencing their future content visibility and recommendation algorithm.
```

## C.7. Prompt for Validation Agent 1

**Prompt for Validation Agent 1**

```
Assess whether the two given natural language instructions convey the same
meaning.  Respond with 'Yes' if they do, or 'No' if they do not.

Please ensure your answer is either "Yes" or "No".

# Instruction A:
{INSTRUCTION A}

# Instruction B:
{INSTRUCTION B}
```

## C.8. Example for Validation Agent 1

**Example for Validation Agent 1**

```
Assess whether the two given natural language instructions convey the same
meaning.  Respond with 'Yes' if they do, or 'No' if they do not.

Please ensure your answer is either "Yes" or "No".

# Instruction A:
For a given list of integers, return a tuple consisting of a sum and a
product of all the integers in a list.  Empty sum should be equal to 0 and
empty product should be equal to 1.

# Instruction B:
In an early disease risk prediction model, develop a function that processes
a list of patient health metrics to calculate comprehensive risk assessment
parameters.  The function should compute two key aggregate indicators:  the
total sum of the patient's health metrics and the cumulative product of
these metrics.  For scenarios with no available health metrics, the sum
should default to 0 and the product should default to 1, ensuring the model
can handle incomplete patient data sets.
```

## C.9. Prompt for Validation Agent 2

**Prompt for Validation Agent 2**

```
Does the following code solve the problem described in the Instruction?
Provide your answer as either 'Yes' or 'No' only.

# Instruction:
{INSTRUCTION}

# Code Solution:
{CODE SOLUTION}
```

## C.10. Example for Validation Agent 2

---

**Example for Validation Agent 2**

```
Does the following code solve the problem described in the Instruction?
Provide your answer as either 'Yes' or 'No' only.

# Instruction:
In a bank's loan risk assessment process, analyze a list of an applicant's
key financial metrics to compute an aggregate financial risk score.
Calculate the total sum of these financial indicators and their cumulative
product to provide a comprehensive risk evaluation metric.  For applicants
with no financial history, the sum should default to 0 and the product
should default to 1, ensuring a standardized risk assessment approach even
for new customers.

# Code Solution:

  from typing import List, Tuple

  def sum_product(numbers: List[int]) -> Tuple[int, int]:
      sum_value = 0
      prod_value = 1
      for n in numbers:
          sum_value += n
          prod_value *= n
      return sum_value, prod_value
```

---

# D. Human Verification

To add an additional layer of validation between the original and `DyCodeEval`-generated prompts, we perform a small-scale manual verification. Given a benchmark dataset and the corresponding generated questions, we randomly sample $N = 30$ problem pairs from each dataset (60 in total), where each pair consists of a benchmark problem and its generated variant. Each pair is independently reviewed by two graduate-level students to assess whether the core algorithm and complexity are preserved. In cases of disagreement, the reviewers discuss the discrepancies until consensus is reached. Out of the 60 reviewed pairs, the annotators initially disagreed on three but were able to resolve all disagreements through discussion, resulting in an overall agreement rate of 95%.

