# OpenReview forum: "DyCodeEval: Dynamic Benchmarking of Reasoning Capabilities in Code Large Language Models Under Data Contamination"
_ICML.cc/2025/Conference — ICML 2025 poster_

### Official Review · Reviewer_TWyo · 2025-03-14

**Overall Recommendation:** 3

**Summary:**

This paper proposes DyCodeEval: a dynamic benchmarking approach designed to evaluate the reasoning capabilities of Large Language Models on code tasks under potential data contamination. By starting with a seed programming problem, DyCodeEval leverages multiple agents to extract and modify problem contexts—without altering the core logic—to generate semantically equivalent variations.

**Claims And Evidence:**

The motivation for handling data contamination is well-founded.

Traditional benchmarks, such as HumanEval and MBPP, may have been seen by models during training; consequently, their results could reflect memorization rather than genuine reasoning ability. The authors’ core idea—using a dynamic process to revise benchmark problems—is promising for mitigating contamination issues.

Still, further elaboration on how they quantitatively measure contamination levels would be helpful. The paper mentions that prior contamination metrics may not align with real-world cases; clarifying the proposed metric or methodology for gauging contamination would strengthen the claim.

**Essential References Not Discussed:**

no

**Experimental Designs Or Analyses:**

The authors suggest that DyCodeEval mitigates data contamination by dynamically generating semantically equivalent but contextually distinct problems.

Beyond final accuracy, incorporating additional metrics (e.g., measuring plan complexity, solution clarity, or partial correctness) could offer a more nuanced view of LLM reasoning skills.

Similarly, comparing performance on newly generated problems that are known to be uncontaminated against older, potentially contaminated benchmarks would illustrate the impact of this approach more concretely.

**Methods And Evaluation Criteria:**

Benchmark Selection: The work primarily focuses on HumanEval and MBPP—two standard code generation datasets that have been around for a while and on which models often perform quite well. It would benefit readers to see experiments on more recent datasets or tasks not so heavily covered in prior training data. Additionally, expanding beyond code completion tasks to more diverse or “harder” datasets could further validate DyCodeEval’s utility.


Plan for More Diversity: While the authors mention multiple agents generating various problem contexts, the paper could better highlight how these newly created tasks genuinely probe the model’s reasoning rather than just superficial text changes. If the final results remain similar in difficulty and performance, additional clarity on whether the plan’s quality improves over iterative changes is needed.

**Other Comments Or Suggestions:**

see above

**Other Strengths And Weaknesses:**

see above

**Questions For Authors:**

see above

**Relation To Broader Scientific Literature:**

Robustness
Code LLM Evaluation
Data Contamination

**Theoretical Claims:**

The proofs follow standard probability and combinatorial arguments.
One potential point for further clarity might be to highlight assumptions of “uniform sampling” in each theorem  (i.e., that scenarios and contexts are chosen with equal probability), as real-world usage could introduce slight biases.

---

> ### Author Rebuttal · Authors · 2025-04-01
>
> Thanks for your valuable comments.
>
> > **How Our Dynamic Metric Mitigates Data Contamination**
>
> For the static metric **pass@k**, the same fixed problem prompt is fed to the LLM multiple times, leveraging its sampling capability to generate different outputs. However, since this prompt is publicly available and remains unchanged, **pass@k** becomes unreliable if the prompt is contaminated in the LLM’s training data.
>
> In contrast, our proposed **DivPass@K** generates multiple randomized problem mutations using our approach before feeding them to the LLM. These dynamically generated prompts are not static, publicly available, or present on the Internet, reducing the risk of data contamination. We will clarify the working mechanism of our dynamic metric in the final version.
>
>
> > **More challenging and uncontaminated benchmarks**
>
> We also applied our approach to LiveCodeBench, a newly collected competition-level programming benchmark sourced from LeetCode and other platforms. The results, shown in the table below, demonstrate that DyCodeEval can be effectively applied to challenging and uncontaminated benchmarks.
> Notably, Qwen2.5-Coder’s accuracy did not drop as significantly as it did from HumanEval to LiveCodeBench. This is because LiveCodeBench was released after Qwen2.5-Coder, reducing the likelihood of data contamination.
>
> |   Model                      | LiveCodeBench | LiveCodeBench +Ours |
> | ---------------------------- | ------------- | ------------------- |
> | CodeLlama-13b-hf             | 21.4         | 18.3               |
> | CodeLlama-7b-hf              | 15.6          | 13.5                |
> | DeepSeek-V2-Lite             | 41.4         | 39.4               |
> | Llama-3.1-8B-Instruct        | 21.3         | 20.8               |
> | Qwen2.5-Coder-7B-Instruct    | 39.4          | 36.5                |
> | deepseek-coder-1.3b-instruct | 22.1          | 19.3                |
> | claude-3.5-haiku             | 59.4         | 60.3               |
> | claude-3.5-sonnet            | 67.7          | 67.6                |
>
>
> > **Uniform sampling in our assumption**
>
> We will revise our theorems and clarify that they rely on the assumption of uniform sampling.
>
>
> > **Other evaluation metrics**
>
>
> Besides the correctness metric, we also consider the test case pass rate to evaluate partial correctness. The results are shown in the following table. We observe that, after applying our transformation, the test case pass rate increases for some models. This is because our transformation generates diverse variants of the problem, which may change the models reasoning and make it poential to get a partial correct solution.
>
> | Model                        | HumanEval | HumanEval + Ours |
> | ---------------------------- | --------- | ---------------- |
> | CodeLlama-13b-hf             | 0.38      | 0.37             |
> | CodeLlama-7b-hf              | 0.29      | 0.33             |
> | DeepSeek-Coder-V2-Lite-Base  | 0.19      | 0.22             |
> | DeepSeek-V2-Lite             | 0.29      | 0.21             |
> | Llama-3.1-8B                 | 0.38      | 0.39             |
> | Llama-3.1-8B-Instruct        | 0.63      | 0.56             |
> | Llama-3.2-1B                 | 0.20      | 0.15             |
> | Llama-3.2-3B                 | 0.31      | 0.33             |
> | Qwen2.5-7B                   | 0.55      | 0.46             |
> | Qwen2.5-7B-Instruct          | 0.63      | 0.56             |
> | Qwen2.5-Coder-7B             | 0.65      | 0.37             |
> | Qwen2.5-Coder-7B-Instruct    | 0.76      | 0.71             |
> | deepseek-coder-1.3b-instruct | 0.54      | 0.43             |
> | claude-3.5-haiku             | 0.86      | 0.78             |
> | claude-3.5-sonnet            | 0.96      | 0.85             |

---

### Official Review · Reviewer_eBWT · 2025-03-15

**Overall Recommendation:** 1

**Summary:**

This paper proposes a framework for augmenting existing coding model evaluation datasets by coming up with new scenarios and contexts to generate semantically similar evaluations. The authors use several LLM steps to produce these questions, and evaluate models while attempting to simulate data contamination. The authors compare their generated evaluation questions against other forms of manipulation, and analyze how the performance of a set of models changes.

**Claims And Evidence:**

In 234-244, the authors claim that their collection of 3 models (Llama 3.2-1B, Llama 3.2-3B, DeepSeek-Coder-1.3B) is diverse "in terms of model architecture, model size, and training methods). This is problematic, as they explore a truly diverse set of models in the following section (12 additional models). Of the 3 original, 2 are the same model family, and all 3 are small models.

Furthermore, the abstract claims that their method creates "semantically equivalent variations", but this does not seem to be validated. The authors claim to perform a human study but do not include details to verify further.

**Essential References Not Discussed:**

N/A

**Experimental Designs Or Analyses:**

Details around finetuning are underspecified. 4.4 does not give details about mutations used for comparison.

All analysis is harmed by the fact that many of the figures have extremely small text, cut off labels, or overlapping content.

The details of the human evaluation are extremely under specified. The appendix simple states "the consistent rate is around 95%" with no further numeric details.

**Methods And Evaluation Criteria:**

The benchmarks used make sense. However, much of the evaluation does not make sense. There are insufficient details on how the authors finetune models to simulate leakage. Fine tuning on a small subset of documents could heavily impact the instruction following capabilities of these models if not considered carefully. Furthermore, training directly on (synthetic) evaluation questions does not simulate pretraining leakage, where implementations might be contained within a larger dataset.

While their capabilities are more limited, there are open data code models that the authors could use for an exact leakage study (Starcoder).

**Other Comments Or Suggestions:**

Typo in the first line of the abstract
763, 765 further typos.
Typos in your prompts 676

**Other Strengths And Weaknesses:**

Strengths:
1. Interesting approach to augmenting evaluations
2. Considers that several of the main evaluations in this area may suffer from leakage

Weaknesses:
1. Analysis is very difficult to understand, compounded by the fact that figures contain extremely small text or overlapping content.
2. Some essential experimental details are not considered (see above)
3. Paper spends substantial content on explaining a series of straightforward LLM prompts as agents.
4. Similarly include sections that do not contribute to the main points (Algorithm 1 is a typical process, section 3.3 does not add to the content). This space would be better used for explaining missing details.
5. Makes claims about semantic equivalence between generated problems but does not sufficiently justify.

**Questions For Authors:**

N/A

**Relation To Broader Scientific Literature:**

This paper studies leakage and analyzes common benchmarks and code models, largely following established literature.

**Theoretical Claims:**

This paper includes unnecessary theorems and claims to analyze the collisions of their method when generating $|S|$ scenarios and $|C|$ coding problem contexts. This analysis and the page of proofs in the appendix appear to be typical statements of balls-and-bins, coupon collector, and hash table collision type problems. These do not add to the substance of the paper except to include more notation and an appendix section. This also does not address the fact that any of the prior LLM steps in this work could simply be repeated or resampled (with temperature, line 252) in the event of a collision.

---

> ### Author Rebuttal · Authors · 2025-04-01
>
> Thanks for reviewing our paper and valuable comments
>
> >**Semantically Equivalent Validation and Human Study**
>
> By “semantically equivalent variations,” we mean that the generated problems can be solved by the same code solution as the original. We validate this through:
>
> 1. **Automated Validation:**
>    A large language model (LLM) acts as a probabilistic oracle to check:
>    - Whether the rewritten problem retains the original meaning.
>    - Whether the original solution remains valid for the new problem.
>
> 2. **Human Verification (Appendix D):**
>    Two graduate students independently reviewed *N = 30* problem pairs per dataset (60 total), assessing whether the core algorithm and complexity were preserved. They initially disagreed on three pairs but reached consensus after discussion, yielding a 95% agreement rate.
>
> We provide the sampled data on our [website](https://github.com/anonymousGithub2022/DyCodeEval/blob/main/resource) (see four CSV files in the directory).
>
>
> >**Finetuning details**
>
> We fine-tune the selected code LLM on randomly sampled portions of the benchmarking dataset, ranging from 25% to 100%, using a standardized instruction tuning objective. The fine-tuning process employs a learning rate of 5e-5, a batch size of 8, and runs for 20,000 steps.
> We acknowledge that fine-tuning on a small subset can impact the instruction-following capabilities of the model due to overfitting. However, this phenomenon is precisely the risk posed by data contamination—overfitted models exhibit artificially high performance on contaminated benchmarks, creating a false sense of intelligence while sacrificing generalizability. This issue is empirically demonstrated in Figure 4 (first row), where the red bars highlight the inflated accuracy due to overfitting, while the blue bars indicate the degradation in general capabilities. The presence of such overfitting underscores the need for contamination-free benchmarking, which is the primary motivation of our work.
> Regarding pretraining leakage, we note that instruction fine-tuning can override or mitigate the effects of pretraining data exposure. Our study does not aim to analyze pretraining leakage directly but rather to simulate its effects in a controlled manner. To achieve this, we follow established methodologies in the literature, where instruction-tuning-stage leakage is widely used to approximate the impact of training data contamination. This approach allows us to systematically examine how leakage-induced overfitting distorts benchmarking results.
>
>
>
>
> >**Unnecessary theorems**
>
> We strongly believe that these theorems are both necessary and valuable. They provide probabilistic guarantees to benchmark providers, ensuring that an entity with ulterior motives cannot easily overfit a model to achieve artificially high scores on our benchmarks. Thanks to our hierarchical transformation framework, we can control the search space at each transformation layer, effectively mitigating the risk of collisions. This approach allows us to maintain a manageable search space at each layer while achieving a significantly large total search space.
>
> The claim that *"prior LLM steps could simply be repeated or resampled (line 252) in case of a collision"* does not apply to our scenario. Benchmark providers keep their scenario and context pools private to prevent manipulation rather than expose them for adversarial exploitation. As these pools act as a private key, our framework ensures transparent benchmarking while minimizing contamination risk. While brute-force overfitting is possible, our theorems show it would require an impractically large number of trials, making it infeasible.
>
> To highlight the advantages of our hierarchical transformation for reducing the rist of collision. We conduct a empirical evaluation, the setup and results are shown on our [website.](https://github.com/anonymousGithub2022/DyCodeEval/tree/main?tab=readme-ov-file#collision-results)
>
>
>
> >**Baselines in Sec4.4**
> We did describe and cite the baseline methods in Section 4.4. Below, we provide further details on the specific mutations:
> - **Token Mutation**: Randomly replaces a token in the original prompt with another token.
> - **Char Mutation**: Randomly inserts a character at a random position in the original prompt.
> - **Func Mutation**: Changes the function name style in the prompt, e.g., renaming "MyMethod" to "my_method."
> - **Insert Line**: Randomly inserts blank lines in the original prompt.
> - **CommSyntax**: Modifies the syntax of comments in the prompt (e.g., changing `# comment` comments to `"""comment"""` style).
> These mutations are derived from the robustness-based mutations proposed by Wang et al. (2023). Additionally, **PPM** (Chen et al., 2024) concatenates the original problem description with a newly defined problem description to test robustness.
> We used publicly available implementations to ensure consistency and reproducibility.

---

### Official Review · Reviewer_oZay · 2025-03-17

**Overall Recommendation:** 4

**Summary:**

The paper presents a method for modifying existing LLM coding benchmarks through a 4-stage pipeline to produce new versions of the benchmark that are unlikely to have appeared in training data. This addresses the challenge that LLM developers face when collecting training data and evaluating their models: that data from their evals may appear in the training corpuses, and removing it is non-trivial. The pipeline consists of a scenario proposer, context generation agent, prompt rewriter, and validation stage. The paper gives theoretical consideration to the possibility of collisions in the task rewrite process. Then they employ the proposed process with two small coding benchmarks; to evaluate it, they consider how model performance changes when different amounts of data are leaked from the benchmark. They also examine the performance of a number of in-the-wild models on the original static benchmark and the new dynamic one, identifying that overfit models struggle on the new benchmark, and hypothesizing that QWEN2.5-CODER-7B may have data contamination. The paper also performs evaluations of diversity of the generated tasks, stability of the benchmark (in spite of its randomness), and evaluates whether weaker but cheaper language models can be used for task generation.
Finally, the paper also introduces a new metric for their dynamic benchmark, DivPass, showing evidence for this metric being more reflective of a models' coding reasoning capabilities than the standard pass@k.

**Claims And Evidence:**

First the paper introduces a method for transforming an existing benchmark (HumanEval and MBPP, two standard, albeit small and "toy" compared to the problems LLMs are commonly used on today, benchmarks for LLM coding capabilities) into a variation of the benchmark unlikely to have been seen in the training data.

It then makes the following claims about the method:
1. First it theoretically places bounds on the likelihood of collisions of the scenario and context of the generated task.
2. Then it measures the effect of various amounts of contamination on the benchmark results, finding the dynamic benchmarking is resistant to contamination. This is the main result of the paper.
3. When benchmarking in the wild models on the new benchmark, the paper reports model performance and finds evidence that Qwen2.5-Coder-7B may be contaminated.

We discuss claim 1 in the theoretical claims section below.
The evidence for claim 2, that the dynamic benchmarking is resistant to contamination, is persuasive: the evidence in Figure 4 is clear. A limitation of this evidence is that the models were merely fine-tuned, not pre-trained, on the contaminated data.
The claim (3) that overfitted models appear as outliers (Figure 5) is reasonable, but there is not hard evidence that Qwen2.5-Coder-7B; the language used for this claim is appropriately couched.

The introduction of the DivPass metric and the measurement of DyCodeEval's stability is a welcome contribution as well. The stability is strong enough (Figure 6) to make this approach trustworthy as a benchmark even when different random tasks are generated at each application.

Finally there is one additional claim that using weaker LLMs for task generation leads to modest degradation in the quality of eval (the consistency rate from the validation stage of generation drops). This is persuasive to me that Haiku is indeed insufficient as a choice of model for task generation, while Sonnet is sufficient (at least for the seed benchmarks selected in the paper).

**Essential References Not Discussed:**

Please also see Is Your Benchmark (Still) Useful? Dynamic Benchmarking for Code Language Models https://arxiv.org/abs/2503.06643, a concurrent work released just in the last few days. (No expectation that it already have been in your paper, of course!)

**Experimental Designs Or Analyses:**

See notes in Claims and Methods sections.

**Methods And Evaluation Criteria:**

The main method of benchmark generation (scenario proposer, context generation, prompt rewriting, and validation) is a sensible way to generate new tasks unseen by the model during training time, even if the original (or a different derived) version of the benchmark leaked into the training set. One limitation of this method concept is that the core solutions (i.e. the algorithmic insights that the solutions lean on) will still have leaked into the training set, but they will be heavily disguised. With today's models, that scenario/context "disguise" may be more significant that with future models.

A limitation of the method is its reliance on a human verification step, which limits the ability to scale up the method.

To evaluate the method the authors measure the stability of the resulting benchmark using HumanEval and MBPP as the seed benchmarks, they measure the benchmark's robustness to (a fine-tuning based approximation of) data contamination, and they measure in the wild performance and speculate about data leakage in real models. They also measure stability to regeneration of the benchmark. These are each quite sensible measures of evaluating the benchmark generation method.
The seed benchmarks are limited in scope.
The robustness to data contamination experiment only uses fine-tuning, not pre-training, which limits the conclusions we can draw.
The true data leakage information about Qwen isn't known, so our ability to measure the benchmark's true leakage-detection ability is limited.
But overall these experiments are compelling, demonstrating that (at least for these simple seed benchmarks and today's models), the approach is sound for generating data contamination resistant variations of the benchmark that are robust to regeneration.

**Other Comments Or Suggestions:**

The pdf repeatedly ran into rendering issues on my machine; my first guess would be that the images in Figure 5 are too large, but I have not investigated further. It might just be an issue on my end. I report this just for your information.

line 12 (the first line) typo: missing space
line 19: remove "to be"
Why is Section 5 titled "Application"? That seems like an oversight.

**Other Strengths And Weaknesses:**

See other sections.

**Questions For Authors:**

Is Sonnet sufficient for applying this task generation procedure to more complex tasks than HumanEval and MBPP?

Is there evidence to validate the speculation that Qwen2.5-Coder-7B contains data contamination (or that the other models do not) beyond that from this benchmark?

**Relation To Broader Scientific Literature:**

There are many code generation benchmarks for LLMs, of which HumanEval and MBPP are two examples. However, the literature routinely recognizes data contamination as a challenge for properly evaluating LLMs.

Generalization or Memorization: Data Contamination and Trustworthy Evaluation for Large Language Models Yihong Dong, Xue Jiang, Huanyu Liu, Zhi Jin, Bin Gu, Mengfei Yang, Ge Li https://arxiv.org/abs/2402.15938

This paper proposes a method to mitigate the challenges of data contamination by using LLMs to modify existing benchmarks.
It stands in contrast to other data contamination mitigation approaches like non-LLM rewrites of existing benchmarks (see next section of review) and manually curated time-cutoff benchmarks like LiveCodeBench.

**Theoretical Claims:**

The theoretical claims are correct given the assumptions stated and the proofs are sound. However, the theorems don't actually tell you what you might think/hope they do on a first read. The theorems correctly bound the probability of a collision between two generated examples provided that the scenario generator generates |S| distinct scenarios, and for each scenario the context phase generates |C| distinct contexts. In generating the |S| scenarios, however, there is the possibility of a collision or near collision (i.e. two very similar scenarios). Similarly when generating contexts conditioned on a scenario, there is the possibility of a collision or near collision. i.e. S or C could contain near-collisions, and the likelihood of this seems more significant that the values given by the bounds.

If indeed we are only concerned with exact collisions, the randomness in the rewriting phase reduces that likelihood considerably.

Are there empirical values that would make sense to show for these bounds? E.g. you could demonstrate what dataset sizes admit what amounts of generated examples without collision.

Note the notation used in theorems 2 and 3 is missing the bars left of S.

---

> ### Author Rebuttal · Authors · 2025-04-01
>
> Thanks for reviewing our paper and valuable comments
>
> ---
> > ​**Empirical values of our theorical bounds**
>
> To empirically evaluate the collision rate of our method, we conduct an experiment on HumanEval. First, we run DyCodeEval on HumanEval to generate an initial set of transformed programming problems. We then repeat this process \( N \) times \( N = 10, 20, 30, 40, 50 \) and measure:
> 1. **Repeat rate** – the proportion of problems from the initial transformed set that reappear in the subsequent \( N \) runs.
> 2. **Collision rate** – the proportion of problems within the \( N \) runs that are duplicates of any previously generated problem, regardless of whether they match the initial set.
>
> To highlight the advantages of our hierarchical transformation, we compare it against a baseline where we prompt an LLM (using the following prompt) to generate a new programming problem from a given seed. We report both the repeat rate and collision rate for this baseline as well.
>
>
> Baseline Prompt
> ```
> Rewrite the following problem description to create a new problem description for new scenario\n\n
>
> Original Problem Description\n
> {ori_inst}\n\n
>
> Please ensure to put your rewritten problem description in <new_problem></new_problem> tags.
> ```
>
> | \# Of Run | Ours           | Ours                   | Baseline       | Baseline               |
> | --------- | -------------- | ---------------------- | -------------- | ---------------------- |
> |           | \# of repeated | repeated rate | \# of repeated | repeated rate |
> | 10        | 0              | 0                      | 3              | 0.018292683            |
> | 20        | 0              | 0                      | 4              | 0.024390244            |
> | 30        | 0              | 0                      | 8              | 0.048780488            |
> | 40        | 0              | 0                      | 9              | 0.054878049            |
> | 50        | 0              | 0                      | 9              | 0.054878049            |
>
> | \# Of Run | Ours            | Ours                    | Baseline        | Baseline                |
> | --------- | --------------- | ----------------------- | --------------- | ----------------------- |
> |           | \# of collision | collision rate | \# of collision | collision rate |
> | 10        | 0               | 0                       | 7               | 0.042682927             |
> | 20        | 0               | 0                       | 17              | 0.103658537             |
> | 30        | 0               | 0                       | 30              | 0.182926829             |
> | 40        | 0               | 0                       | 36              | 0.219512195             |
> | 50        | 0               | 0                       | 39              | 0.237804878             |
>
>
> ---
> > ​**Is Sonnet sufficient for more complex tasks**
>
> To assess whether Sonnet is sufficient for more complex tasks, we apply our approach to LiveCodeBench, a **competition-level** programming benchmark. The following table shows the number of tokens in these three datasets.
> For LiveCodeBench, we randomly selected 100 seed and transformed programming problem pairs and evaluated their semantic equivalence. Our analysis found that 92 out of 100 pairs were semantically equivalent, demonstrating the effectiveness of our transformation approach.
>
> | Dataset       | Min | Avg.  | Max |
> | ------------- | --- | ----- | --- |
> | LiveCodeBench | 54  | 242.7 | 693 |
> | HumanEval     | 4   | 55.6  | 430 |
> | MBPP          | 7   | 16    | 47  |
>
>
>
> ---
> > ​**Other evidence to validate that Qwen2.5-Coder-7B contains data contamination**
>
> Another indication that Qwen2.5-Coder-7B may be overfitted comes from LiveCodeBench, where its evaluation also shows an unusually large accuracy drop on newly collected programming benchmarks, similar to our findings in Figure 6.
>
> ---
> > ​**Essential References Not Discussed**
>
> We will add all mentioned paper to related work.

---

### Official Review · Reviewer_3wZN · 2025-03-17

**Overall Recommendation:** 1

**Summary:**

This paper introduces a novel code LLM benchmark that leverages metamorphic testing to address challenges associated with current benchmarks' reliance on publicly available, human-curated datasets.

**Claims And Evidence:**

Yes

**Essential References Not Discussed:**

No

**Experimental Designs Or Analyses:**

Yes

**Methods And Evaluation Criteria:**

Yes

**Other Comments Or Suggestions:**

Many presentation details require improvement. For example, the images in Figure 6 appear truncated, which impacts readability.

**Other Strengths And Weaknesses:**

1. I have concerns about the benchmark's discriminative power. For instance, in the right subplot of Figure 5 (MBPP), only a few models appear as outliers. Additionally, open-source models seem to perform as well as closed-source models on the new benchmark, which raises doubts about whether the benchmark's difficulty is sufficient to track future model advancements.
2. The paper lacks references to highly relevant works, such as Li et al. "EvoCodeBench: An Evolving Code Generation Benchmark with Domain-Specific Evaluations" (NeurIPS 2024) . The existence of such closely related work significantly undermines the claimed novelty.

**Questions For Authors:**

Please see `Other Strengths And Weaknesses`.

**Relation To Broader Scientific Literature:**

No

**Theoretical Claims:**

None

---

> ### Author Rebuttal · Authors · 2025-04-01
>
> Thanks for reviewing our paper and valuable comments.
>
> > ​**Concern about the benchmark's discriminative power**
>
> We appreciate the reviewer’s feedback and the opportunity to clarify our findings. However, we believe there may be some misunderstandings regarding Figure 5.
>
> First, the lower number of outliers in Figure 5 does not imply that our benchmark lacks discriminative power. The key evidence for distinguishing overfitted models is in Figure 4, not Figure 5. In Figure 4, we fine-tune models with controlled contamination and observe a significant accuracy drop on our benchmark as leakage increases (second row). This confirms that models overfitted to contaminated data perform well on the original benchmark but fail on ours.
>
> In contrast, Figure 5 evaluates potential data contamination in publicly available LLMs, not discriminative power. Without access to these models’ training data, we cannot confirm overfitting but instead analyze their performance differences. Our findings show:
> 1) A linear relationship between accuracy on our benchmark and the original, indicating comparable problem complexity.
> 2) Anomalous behavior in certain models, such as Qwen2.5-Coder-7B, which experiences an unusually large accuracy drop, falling outside the 95% confidence region. While we cannot confirm contamination, this suggests potential data leakage, which is why we use the term “may be contaminated.”
>
> Second, the fact that open-source models perform similarly to closed-source ones does not mean our benchmark is too easy to track future advancements. Our focus is on *transparent* evaluation rather than increased difficulty. A reliable benchmark should measure true generalization while avoiding misleading performance inflation caused by data contamination.
>
>
>
> > **Relationship with EvoCodeBench**
>
> EvoCodeBench is constructed by collecting programming problems from GitHub, following a similar approach to LiveCodeBench, as discussed in our *Introduction* section. However, this method has several limitations: (1) It shifts the burden of manual question design to coding platform authors. (2) Since problems come from public GitHub repositories, existing models may have already seen them, raising concerns about data contamination. (3) EvoCodeBench relies on external contributions, leading to infrequent updates—the latest update, per their [website](https://huggingface.co/datasets/LJ0815/EvoCodeBench/tree/main), was nine months ago, which is inadequate given the rapid pace of model development.
>
> While DyCodeEval is fundamentally different, we acknowledge EvoCodeBench as related work and will include it in the related work section. However, its existence does not diminish our contributions, as DyCodeEval is fully automated and scalable.
>
>
> > **Figure 6**
>
> We have revised Figure 6, on our [website.](https://github.com/anonymousGithub2022/DyCodeEval/tree/main?tab=readme-ov-file#stability-of-dycodeeval)

---

### Decision · Program_Chairs · 2025-05-01

**Decision:**

Accept (poster)

**Comment:**

The manuscript proposes a dynamic benchmarking strategy for evaluating the reasoning capabilities of Large Language Models on code tasks under potential data contamination. The method relies on a seed dataset/programming problem, from which it generates semantically equivalent variations. It does so by instructing LLMs to extract and modify problem contexts without altering the core logic. Evaluation using 2 standard code generation datasets (HumanEval & MBPP) on several state-of-the-art code LMs in the 7B range shows the merits of the proposed approach over static evaluation strategy where contamination is a key issue.

The motivation of the work is clear, the work is well-founded, and makes an attempt in the right direction to help robust evaluation of large language models (coding domain in this case). With all benchmark papers in the rapidly evolving LLM landscape, we can consider two questions: (1) Does the proposed process/strategy create more difficult questions than the existing questions to challen, rather than attempting to create just another dataset? (2) What value does the benchmark add over existing ones, in terms of utility, metrics, diversity, maintenance, etc. I think the paper answers both these questions satisfactorily.

The reviewers point out similar approaches for dataset creation in the recent literature, but the authors' rebuttal alleviates serious concerns. The two reviewers remain positive after the rebuttal, and the authors have satisfactorily addressed the concerns raised by the two negative reviewers.

Therefore, I recommend accept. I request the authors to incorporate the comments in the final version (for instance, avoiding strong claims like 'semantic equivalence' without formal statements).